# Systemic Lidocaine Infusion for Post-Operative Analgesia in Children Undergoing Laparoscopic Inguinal Hernia Repair: A Randomized Double-Blind Controlled Trial

**DOI:** 10.3390/jcm8112014

**Published:** 2019-11-18

**Authors:** Hye-Mi Lee, Kwan-Woong Choi, Hyo-Jin Byon, Ji-Min Lee, Jeong-Rim Lee

**Affiliations:** 1Department of Anesthesiology and Pain Medicine, Yonsei University College of Medicine, 50-1 Yonsei-ro, Seodaemun-gu, Seoul 03722, Korea; ham9906@yuhs.ac (H.-M.L.); jinoben@yuhs.ac (H.-J.B.); leejm105@yuhs.ac (J.-M.L.); 2Anesthesia and Pain Research Institute, Yonsei University College of Medicine, 50-1 Yonsei-ro, Seodaemun-gu, Seoul 03722, Korea; 3Department of Anesthesiology and Pain Medicine, National Health Insurance Service Il San Hospital, 100 Ilsan-ro, Ilsandong-gu, Goyang-si, Gyeonggi-do 10444, Korea; sanctum@nhimc.or.kr

**Keywords:** analgesia, general anesthesia, inguinal hernia, intravenous, laparoscopy, lidocaine, pediatric, post-operative pain

## Abstract

Systemic lidocaine can provide satisfactory post-operative analgesia in adults. In this study, we assessed whether intravenous lidocaine is effective for post-operative analgesia and recovery in children undergoing laparoscopic inguinal hernia repair. A total of 66 children aged from six months to less than six years were classified in either the lidocaine (L) or control (C) groups. Children in Group L received a lidocaine infusion (a bolus dose of 1 mL kg^−1^, followed by a 1.5 mg kg^−1^ h^−1^ infusion), whereas Group C received the same volume of 0.9% saline. The primary outcome was the number of patients who presented face, legs, activity, crying and consolability (FLACC) scores of four or more, and therefore received rescue analgesia in the post-anesthesia recovery care unit (PACU). Secondary outcomes included the highest FLACC score in the PACU, FLACC, and the parents’ postoperative pain measure (PPPM) score at 48 h post-operation, as well as side effects. The number of children who received rescue analgesia in the PACU was 15 (50%) in Group L and 22 (73%) in Group C (*p* = 0.063). However, the highest FLACC score in PACU was lower in Group L (3.8 ± 2.4) than in Group C (5.3 ± 2.7) (*p* = 0.029). In conclusion, systemic lidocaine did not reduce the number of children who received rescue analgesia in PACU.

## 1. Introduction

Systemic lidocaine can provide satisfactory analgesia with less opioid use and enhance the overall quality of recovery, particularly after open or laparoscopic abdominal surgery in adults [1,2]. Despite such advantages, systemic lidocaine has not been studied thoroughly in the pediatric population.

Laparoscopic approaches in pediatric surgery have increased significantly [3], but some studies indicate that laparoscopic hernia repair is not superior to open surgery in terms of the severity of post-operative pain [4,5]. As an example, if an adequate analgesic protocol was not applied, then the median face, legs, activity, crying and consolability (FLACC) score was still five (95% confidence interval (CI) 3.97 to 6.03) two hours after laparoscopic inguinal hernia repair in toddlers [6]. Pain from laparoscopic surgery is caused by both somatic- and pneumoperitoneum-induced visceral pain [7]. Therefore, although regional analgesia affects somatic pain, it is unlikely to fully alleviate the response from visceral stimulation [8]. Accordingly, convincing systemic analgesic methods need to be further studied.

We hypothesized that systemic lidocaine would provide reliable analgesia in children undergoing laparoscopic hernia repair surgery. We focused on how many children would need rescue opioid analgesia after surgery and whether systemic lidocaine could reduce this number. Accordingly, we assessed post-operative pain in children using the FLACC scale at the post-anesthesia recovery care unit (PACU), and rescue opioid was administered if the FLACC score was four or greater [9]. The primary outcome was the number of patients who received rescue analgesia in PACU. The analgesic effect of lidocaine was evaluated using the FLACC score and the parents’ postoperative pain measure (PPPM) score 48 h after surgery.

## 2. Materials and Methods

### 2.1. Ethics

Ethical approval for this study (Ethical Committee #4-2013-0692) was provided by the Institutional Review Board of Severance Hospital, Seoul, Republic of Korea on 29 November 2013. This study was registered at ClinicalTrial.gov (NCT02007330, 10 December 2013). Written informed consent was obtained from the parents of all children.

### 2.2. Patients

Children aged 6 months to <6 years, who had an ASA (American Society of Anesthesiologists) physical status of 1 or 2 and were scheduled for elective laparoscopic inguinal hernia repair surgery between December 2013 and June 2015, were enrolled in this study. Patients were excluded if they had clinical evidence of cardiopulmonary, renal, or hepatic disease; cerebral dysfunction; or neurological disease. In addition, children who had been taking analgesia within 2 weeks of the date of surgery, with a history of respiratory infection during the preceding 2 weeks, or those with a known allergy to lidocaine, were excluded.

### 2.3. Anesthesia Protocol

General anesthesia followed the conventional pediatric anesthesia protocol of Severance Hospital (Seoul, Republic of Korea). According to the hospital’s policy, patients were admitted early on the morning of the day of surgery, and intravenous catheterization with a 24 gauge angiocatheter was performed in the ward. When the patient arrived at the pre-operative treatment room, the attending anesthesiologist checked their medical history and undertook a physical examination to check for exclusion criteria. Because only the patient is allowed to enter the operating room in our institution, the patient was sedated in the pre-treatment room before being separated from their caregiver. Intravenous atropine (0.01 mg kg^−1^), lidocaine (1 mg kg^−1^), and Propofol (2–3 mg kg^−1^) were administered in the pre-treatment room. As soon as the child was sedated, they were transferred to the operating room quickly, and the attending anesthesiologist started assisted mask ventilation with 100% oxygen and 3%–4% sevoflurane; routine monitoring, including pulse oximetry, capnography, electrocardiography, and non-invasive blood pressure measurements, was conducted simultaneously. After confirming that there was no eyelash reflex or other signs of consciousness, rocuronium (0.4–0.6 mg kg^−1^) and fentanyl (1 µg kg^−1^) were administered. Then, after 2–3 min, endotracheal intubation was performed. Anesthesia was maintained with 0.8–1.2 MAC (minimal alveolar concentration) of sevoflurane in a mixture of 40% oxygen. Ventilation targeted an end-tidal carbon dioxide concentration of 4.7–5.3 kPa, with a delivered tidal volume of 6–8 mL kg^−1^.

The surgical procedures were performed with three trocars. After the trocar had been introduced, a pneumoperitoneum was created with 10 mmHg intraabdominal pressure. The surgeon approached the internal ring level and ligated it with a non-absorbable purse string suture. When the trocars were removed, fentanyl (0.5 µg kg^−1^) was administered for post-operative analgesia. At the end of surgery, sevoflurane was discontinued, atropine (0.01 mg kg^−1^) and neostigmine (0.02 mg kg^−1^) were administered for the reversal of residual muscle relaxation, and the child was ventilated with 100% oxygen at 6 L min^−1^. Extubation was performed when the patient presented all the following signs: grimace, eye opening, crying face, spontaneous turning of the head, and purposeful movement of limbs. 

The patient was transferred to the PACU, where standard monitoring was applied, and vital signs were checked every 10 min by nurses. If the children experienced separation anxiety, the parents could stay with the patients during recovery. After 30 min of recovery, the nurses checked the Modified Aldrete score, which included the patients’ respiration, SpO_2_, mental status, circulation, and reflex ability. If the score was 9 or higher, the patients were discharged from the PACU according to the doctor’s instructions.

### 2.4. Studying the Drug Administration Protocol

Randomization was performed by the principal investigator. Patients were assigned to either the lidocaine group (Group L) or the control group (Group C), according to the randomization table provided. When the patient arrived in the pre-treatment room, the principal investigator prepared the drug according to group allocation and provided it to the attending anesthesiologist, who was unaware of its contents. Children in both groups were administered 1% lidocaine 1 mg kg^−1^, over 1 min immediately prior to the administration of Propofol, to attenuate the pain upon injection of Propofol. Thereafter, patients in Group L received a continuous infusion of 1% lidocaine at a rate of 1.5 mg kg^−1^ h^−1^, which was started intraoperatively prior to incision and continued at least until extubation. Patients in Group C received a similar volume of 0.9% saline over the same time period. At the end of surgery, fentanyl (0.5 µg kg^−1^) was administered to both groups.

### 2.5. Outcome: Post-Operative Pain Measurement and Management

In the PACU, the severity of post-operative pain was assessed using the FLACC scale three times (10 and 20 min after admission and before discharge) by one designated researcher, who was unaware of the group allocation. If the FLACC score was 4 or higher, fentanyl (0.5 µg kg^−1^) was administered as rescue analgesia [10]. If the FLACC score was 7 or higher, ketorolac (1 mg kg^−1^, maximum 30 mg) as a double rescue analgesia was additionally administered.

The post-operative pain score in the ward was also evaluated at 4, 8, 12, and 24 h after surgery using the FLACC scale. To assess any pain after discharge, the researcher obtained the PPPM score from the guardian by telephone on the day after discharge (about 48 h after surgery) [10].

The primary outcome was the number of children who received rescue analgesia in PACU. Secondary outcomes included the highest score for FLACC in PACU, the FLACC scores at 4, 8, 12, and 24 h after operation, and the PPPM score about 48 h after surgery. Other adverse events (nausea, vomiting, seizure-like abnormal movement, and arrhythmia) were also noted. The latter two could be indicative of lidocaine toxicity.

Our sample size was calculated from data obtained in a previous study, which demonstrated that the incidence of moderate to severe pain in children undergoing laparoscopic appendectomy was 80% [11]. We assumed that the incidence of patients presenting with a FLACC score of 4 or more after surgery would be reduced from 80% to 50% by administering lidocaine. Fisher’s exact test showed that the required sample size would be 30 patients in each group, with a significance level of 5% and a power of 90%. A final sample size of 33 children per group was selected, to allow for a dropout rate of 10%.

### 2.6. Statistical Analysis

The statistical analysis was performed using SPSS 20.0 (SPSS Inc., Chicago, IL, USA). Univariate statistical analyses were conducted to analyze the baseline characteristics. Under the assumption of normal distribution, we used Shapiro–Wilk tests. According to the normality of the data, continuous variables (age, height, weight, duration of operation, duration of anesthesia, and the highest FLACC score) were analyzed using Student’s *t*-test or the Mann–Whitney U test, and are reported as the mean ± standard deviation or median (interquartile range, IQR). All categorical and ranking variables (sex and ASA physical status) were analyzed using the χ^2^ test or Fisher’s exact test, and are expressed as *n* (%).

For primary outcome analysis, categorical variables are expressed as *n* (%) and analyzed by an χ^2^ test. *p* <0.05 was considered statistically significant. For the secondary outcome analysis of the highest FLACC scores in PACU, we analyzed the FLACC scores using Student’s t-test, and expressed the results as the mean ± SD. Because the FLACC scores in the ward were measured at four different times for the same patient, a Bonferroni correction for four comparisons was used, and *p* <0.012 was considered to be statistically significant. Serial data were analyzed using a linear mixed model with unstructured covariance, with fixed effects including time, group, and interactions, as well as with a random effect on the response variable. The PPPM score was analyzed using a Mann–Whitney U test, and adverse events were analyzed using the χ^2^ test. A *p*-value <0.05 was considered statistically significant.

## 3. Results

Among the 66 children included in the study, 30 patients in Group L and 30 in Group C were finally enrolled. Six patients dropped out of the study, and two patients in Group L and three patients in Group C were withdrawn, as their parents refused monitoring 48 h after surgery. One patient in Group L underwent re-operation due to a surgical issue (Figure 1).

The baseline characteristics of the patients included in the study are listed in Table 1.

### 3.1. Primary Outcome Parameters

A total of 15 children (50%) received rescue analgesia in the recovery room in Group L and 22 (73%) in Group C (*p* = 0.063; Table 2). The number of children that received double rescue analgesia was significantly higher in Group C than Group L (*p* = 0.002; Table 2).

### 3.2. Secondary Outcome Parameters

The highest FLACC score during the stay in PACU differed between Group C (5.3 ± 2.7) and Group L (3.8 ± 2.4; *p* = 0.029).

FLACC scores in Group L were lower than those in Group C at 4, 8, 12, and 24 h after operation (Figure 2). When a linear mixed model analysis was used to analyze the relationship between time and pain, a statistically significant difference was found in the degree of improvement in the pain score between the two groups over time (Bonferroni corrected *p* = 0.010). Even after discharge, the patients in Group L had a significantly lower PPPM score than those in Group C (median 1.0, IQR [0.0 to 1.0] vs. 3.0 [2.0 to 4.0], *p* < 0.001; Figure 3).

None of the cases reported adverse effects, such as nausea, vomiting, seizure, or arrhythmia, after using lidocaine.

## 4. Discussion

Our findings demonstrate that intravenous lidocaine infusion reduced the severity of the pain measured by the FLACC score in children undergoing laparoscopic hernia repair, although the number of patients with FLACC scores of four or higher who received rescue analgesia was not significantly reduced. Along with this outcome, children in Group L had lower FLACC and PPPM scores throughout the post-operative period until 48 h post-operation.

Unlike in adults, lidocaine intravenous infusion in children has not been extensively studied. So far, only two randomized controlled studies have been published. In one study, a lidocaine bolus dose of 1.5 mg kg^−1^, followed by a continuous infusion at a rate of 1.5 mg kg^−1^ h^−1^ for six hours, was administered to children undergoing major abdominal surgery. This treatment attenuated the increase in serum cortisol levels, reduced daily fentanyl requirements, hastened the return of bowel functions, and reduced the length of the hospital stay for the children [12]. The other published study indicated that lidocaine (1.5 mg kg^−1^ over five minutes followed by 2 mg kg^−1^ h^−1^) decreased postoperative vomiting in children undergoing an elective tonsillectomy [13]. The serum’s lidocaine concentration was measured in both studies, and in no cases were toxic plasma concentration, neurological disturbances (seizures, numbness, tingling, or paresthesia), or cardiovascular collapses detected in any of the participants [12,14]. University Children’s Hospital Zurich, Switzerland implemented the protocol of intravenous lidocaine infusion for children undergoing laparoscopic surgery, and adverse effects were not reported [13]. According to the Cochrane analysis, in adults undergoing open or laparoscopic abdominal surgery, a continuous lidocaine infusion rate of 1.5 mg kg^−1^ h^−1^ reduced pain immediately after, and until 24 h, with no increased risk of adverse effects such as death, arrhythmias or signs of lidocaine toxicity [15]. Considering both the effectiveness and safety of these previous trials, we used a bolus injection of 1 mg kg^−1^, followed by a continuous infusion of 1.5 mg kg^−1^ h^−1^.

According to our results, the administration of lidocaine for about 60 min statistically lowered the pain score (mean FLACC 3.8 in Group L vs. 5.3 in Group C) immediately after surgery and for up to two days after surgery. However, since rescue analgesia was to be administered if the FLACC score was four or greater [9], the number of patients who needed rescue analgesia in the PACU was not significantly reduced by lidocaine administration. One possible reason for this is lidocaine’s mechanism of action. Through various mechanisms, systemic lidocaine is thought to prevent central sensitization, as well as spinal or peripheral hypersensitivity in response to nociceptive surgical stimuli, along with wound healing, and anti-thrombotic and anti-inflammatory effects [16,17]. In other words, the effects of lidocaine on post-operative pain reduction are assumed to result from indirect mechanisms. Therefore, lidocaine itself may not be sufficient as a sole analgesic immediately after operation. When systemic lidocaine is incorporated into one element of multimodal analgesia, its role in reducing the severity of pain, and its prolonged analgesic effects, will be augmented if combined with any modalities or drugs for acute postoperative pain control. 

In this study, we focused on how many children would need rescue opioid analgesia after surgery, as well as whether systemic lidocaine could reduce the number of patients who received a rescue opioid. In children, pain control is required if the FLACC score is four or greater after surgery. Contrary to the VAS (visual analogue scale) for pain, no guideline exists for the minimal differences in FLACC scores that signify clinical importance. Therefore, it was difficult to define how much of a reduction in the FLACC score would be statistically and clinically significant. This is why our primary outcome was defined as the number of patients with FLACC scores of four or more points.

Some adult studies have demonstrated that systemic lidocaine can enhance the quality of recovery [18,19]. Since no verified scoring system for this exists for children, such as the quality of recovery-40 (QoR-40) for adults, we used a PPPM scale to assess pain as well as quality of recovery in children [20]. PPPM applies a pain threshold scale that uses a yes (score of 1) and no (score of 0) system to assess daily routines or children’s behavior, and the questions ask whether a child’s behavior has changed compared with before surgery. According to our results, the mean PPPM score was 1.3 in the lidocaine group, which was significantly different from the 3.2 in the control group. It is debatable if this statistical difference correlates to clinical significance in terms of pain, since a PPPM score of six or higher is considered to be significant pain. However, from the parent’s perspective, 48 h after surgery, only 5 out of 30 children in the lidocaine group, compared to 19 out of 30 children in the control group, showed behavioral changes in three or more questions. Therefore, as it does for adult patients, systemic lidocaine is likely to help children return to their daily activities and improve their quality of life after procedures that require anesthesia and surgery.

Apart from the doses used, the limitations of this study include safety issues and the pain scale used. First, we only included children aged six months to less than six years, due to the concerns of possible systemic toxicity for patients younger than six months of age. We could not include children aged six years or older because the FLACC scale is a tool for assessing pain in children under seven years of age. Second, the analgesic protocol in the ward and at home was not standardized. However, almost all children presented pain lower than the threshold that required treatment (FLACC ≥4 and PPPM ≥6) in the ward or after discharge. Third, since laparoscopic hernia surgery is a minor surgical procedure, the PPPM score for evaluating long-term post-discharge recovery was quite low in both groups, and the clinical implications are questionable. Therefore, we recommend studying the recovery effect of lidocaine on major operations with moderate to severe pain after surgery.

In conclusion, systemic lidocaine reduced the severity of pain after laparoscopic hernia repair in toddlers, although the degree of reduction was not enough to eliminate their need for rescue analgesia. The possible beneficial analgesic effects of systemic lidocaine persisted for up to 48 h in children undergoing laparoscopic hernia repair.

## Figures and Tables

**Figure 1 jcm-08-02014-f001:**
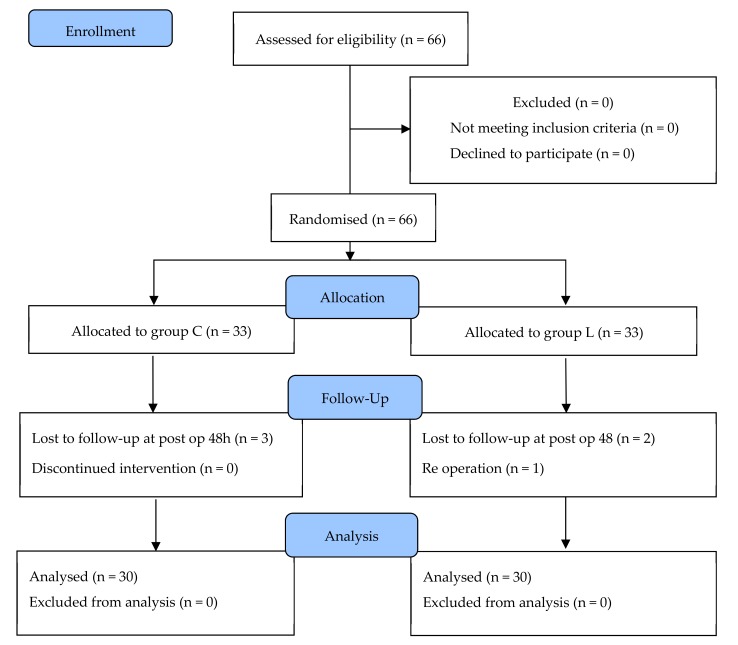
Flow diagram of the study.

**Figure 2 jcm-08-02014-f002:**
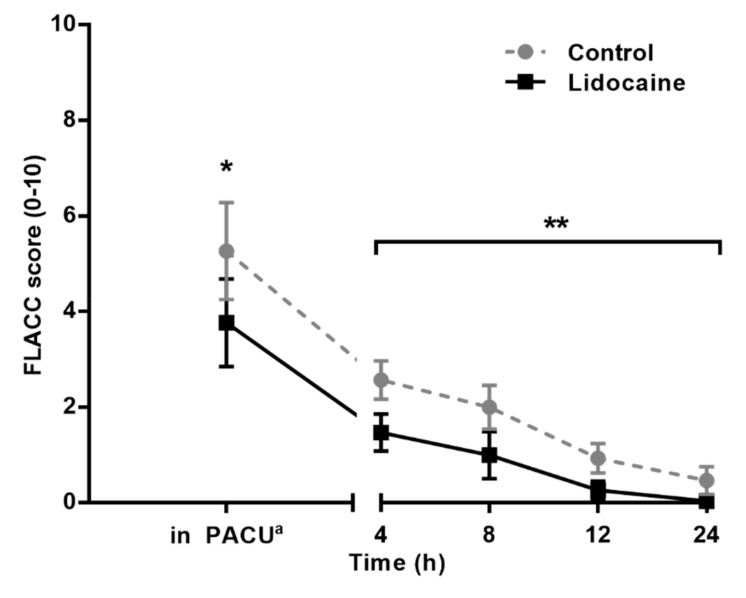
Face, legs, activity, crying, consolability (FLACC) scores in the PACU and in the ward. Values are shown as mean, and the bars indicate the 95% confidence interval for the mean. * *p =* 0.017. ** Bonferroni corrected *p =* 0.010. ^a^ The highest FLACC score in PACU.

**Figure 3 jcm-08-02014-f003:**
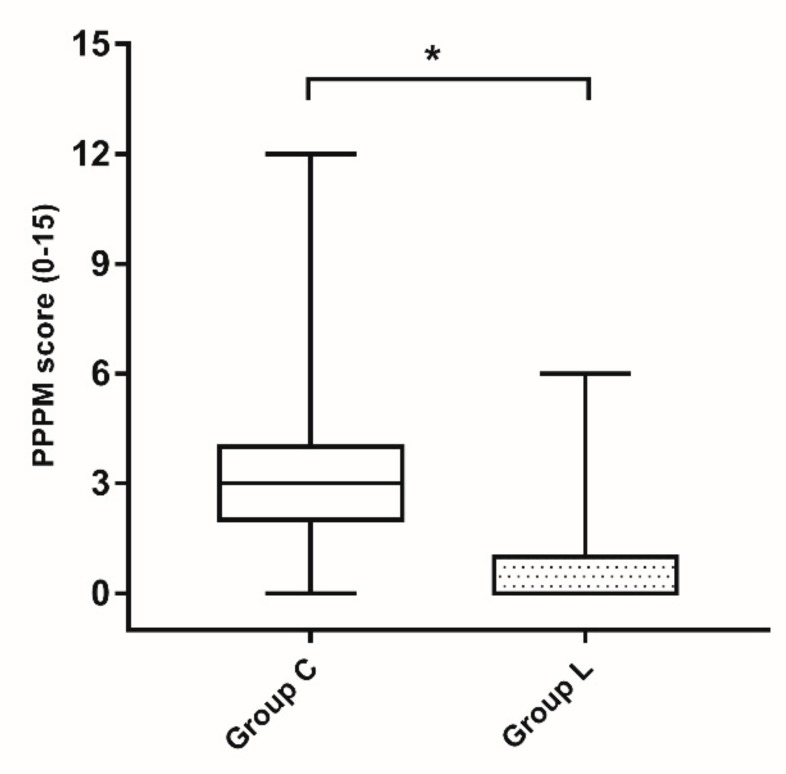
Post-operative pain measurement (PPPM) scores completed by the parents 24 h after discharge. Boxes represent the interquartile range; black bars in boxes represent the median. The ends of vertical lines indicate the minimum and maximum data values. * *p* < 0.001

**Table 1 jcm-08-02014-t001:** Baseline characteristics of the patients included in the two study groups.

Variable	Group C	Group L
	*n* = 30	*n* = 30
Sex (male)	20 (66.7%)	20 (66.7%)
Age (months)	36.0 [25.0 to 47.0]	36.0 [22.5 to 49.5]
Height (cm)	96.2 [87.6 to 105.0]	98.0 [89.3 to 106.8]
Weight (kg)	14.5 [12.3 to 16.8]	15.0 [12.4 to 17.7]
ASA* physical status; class 1	28 (93.3%)	29 (96.7%)
Duration of operation (min)	35.0 [31.0 to 39.0]	40.0 [30.0 to 50.0]
Duration of anesthesia (min)	60.0 [55.0 to 65.0]	65.0 [55.0 to 75.0]

Note: Values are expressed as the median [interquartile range] or number (%). * ASA; American Society of Anesthesiologists classification.

**Table 2 jcm-08-02014-t002:** The number of patients who received rescue analgesia in the post-anesthesia care unit (PACU).

	Group C *n* = 30	Group L *n* = 30	*p* Value
No rescue analgesia	8 (26.7%)	15 (50.0%)	
Rescue analgesia	22 (73.3%)	15 (50.0%)	0.063
Single rescue analgesia	7 (23.3%)	10 (33.3%)	
Double rescue analgesia	15 (50.0%)	4 (13.3%)	0.002 *

Note: Values are expressed as the number (%). * *p* value was <0.05.

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
