# Peer review of "Systemic Lidocaine Infusion for Post-Operative Analgesia in Children Undergoing Laparoscopic Inguinal Hernia Repair: A Randomized Double-Blind Controlled Trial"

_jcm, 2019, doi:10.3390/jcm8112014_

Round 1
Reviewer 1 Report
In their manuscript, entitled “Systemic Lidocaine Infusion for Post-Operative Analgesia in Children Undergoing Laparoscopic Inguinal Hernia Repair: A Randomised Double-Blind Controlled Trial”, the authors are presenting the results of a clinical study evaluating the effect of intravenous lidocaine on post-operative pain after laparoscopic inguinal hernia surgery in children.
The overall idea of the study is interesting. However, there some issues within this manuscript, which I would recommend the authors to address:
Major:
Results, Figure 1: This important finding would possibly be more comprehensible if presented as a table. Results: Is there any information available, how many children received both fentanyl and ketorolac in both groups? This comparison might be another interesting point to add. Discussion, p6: The results of the Cochrane analysis should be discussed in more detail. Unfortunately, lidocaine does not seem to be as “effective” (line 200), as we initially thought. General: Unfortunately, the authors did not apply any baseline analgesia, e.g. metamizol or paracetamol, to their patients, which should not only be standard for such procedures but it might have also changed their results. This should be part of the discussion as well. General: As this is an RCT, a CONSORT flow chart should definitely be part of the manuscript.Minor:
There are some minor linguistic issues throughout the manuscript, which can and should be resolved easily.Author Response
Major
Results, Figure 1: This important finding would possibly be more comprehensible if presented as a table.
I fully agree with your comment. We revised Figure 2 to Table 2.
Results: Is there any information available, how many children received both fentanyl and ketorolac in both groups? This comparison might be another interesting point to add.
According to your opinion, we added the number of children who received double rescue analgesics in Table 2 (line164-168).
Discussion, p6: The results of the Cochrane analysis should be discussed in more detail. Unfortunately, lidocaine does not seem to be as “effective” (line 200), as we initially thought.
We rewrite the results of the Cochrane analysis in discussion section. (line208-211) We also regret that the results are not positive enough. The reasons why we could not achieve the effect we expected were 1) the operation was too short and 2) intravenous lidocaine was not sufficient as principal or sole analgesia.
General: Unfortunately, the authors did not apply any baseline analgesia, e.g. metamizol or paracetamol, to their patients, which should not only be standard for such procedures but it might have also changed their results. This should be part of the discussion as well.
As we already mentioned in the method section (line 88-89); “When the trocars were removed, fentanyl (0.5 µg kg−1) was administered for post-operative analgesia.”, we applied fentanyl as a baseline analgesia. In addition, the pediatric general surgeons in our institute usually don’t prescribe any analgesics at hospital discharge because the children’s pain score is supposed to be low. Also, most parents have paracetamol or NSAID at home for cold medicine or fever remedy, so the surgeons thought even if they prescribed analgesics, the medicine could be discarded. Therefore, we educated the parents to give acetaminophen or NSAID they have at home to children if the children presented any signs of pain. We had considered gathering this information and documenting oral medications given at home. However, home analgesics were not uniformed so it was very difficult to compare them without having standardized dosage scheme. We mentioned about this issue in the limitation paragraph, “Second, the analgesic protocol in the ward and at home was not standardized. However, almost all children presented pain less than the threshold, which required treatment (FLACC ≥ 4 and PPPM ≥ 6) at ward or after discharge.” (line251)
General: As this is an RCT, a CONSORT flow chart should definitely be part of the manuscript.
The Consort flow chart was updated in the manuscript, Figure 1. (line157)
Minor:
There are some minor linguistic issues throughout the manuscript, which can and should be resolved easily.
We edited the manuscript using MDPI English editor.

Reviewer 2 Report
This is the first review for MS# jcm-600438
The authors present a paper on IV lidocaine infusions vs placebo in children undergoing lap inguinal hernia repair
The entire manuscript needs detailed revision by a native English speaker, mostly for syntax and and sentence structure.
While I think the overall paper is of interest, there are significant flaws with respect to methodology that require addressing.
1) Does the institution routinely admit ASA 1 and 2 patients for 24 hrs post an uncomplicated hernia repair? If not, the authors should justify why this was done?
2) All post pacu data such as the Flacc and PPPM is of no value- there is no data on post op pain regiments after pacu and these analgesics were not controlled.
3) was any local anesth used for a block caudal or nerve block?
4) the authors should state in 2.4 exactly when the lidocaine infusion was started and stopped
5) the most important problem with this study comes with the power calculation- the authors used a seemingly inappropriate study and comparision for this- Lap appys is not a good comparison as they are not elective and associated with significant preemptive pain. not to mention that the source of this paper is highly obscure. (possibly this data could be used to power a reasonable sample size)
6) I dont think that with the subjective FLACC in pacu that a score difference of 3.8 to 5.3 has any clinical relevence- this is also bolstered by the fact that their treatment of pain was no difference
7) section 4 page 6 ln 200- the comment on neuropathic pain is not relevant here and should be removed.
Author Response
Thank you for your review. We edited the manuscript using MDPI English editor.
1) Does the institution routinely admit ASA 1 and 2 patients for 24 hrs post an uncomplicated hernia repair? If not, the authors should justify why this was done?
When this study was conducted, all children stayed overnight at the hospital due to the surgeon’s policy. Therefore, FLACC measurement was possible until 24 hours after surgery. Currently, most inguinal hernia surgery in children at out hospital is also performed as day-stay surgery.
2) All post pacu data such as the Flacc and PPPM is of no value- there is no data on post op pain regiments after pacu and these analgesics were not controlled.
Thank you for your comment. We described this in the discussion as follows: Second, the analgesic protocol in the ward and at home was not standardized. However, almost all children presented pain lower than the threshold that required treatment (FLACC ≥ 4 and PPPM ≥ 6) in the ward or after discharge. (line251)
3) was any local anesth used for a block caudal or nerve block?
Neither local infiltration nor nerve blocks were applied. Local or regional block could help to provide better analgesia, but our study was designed to focus on the effect of systemic lidocaine, and local infiltration or nerve blocks would make the interpretation of the effect of systemic lidocaine difficult. In addition, if local or regional block was conducted, local anesthetic used for local or regional block would be eventually absorbed systemically and lidocaine concentration would be higher than expected. In the case of the caudal block, we do not do it because it can exacerbate the rise of the ICP by the pneumoperitoneum [1,2].
4) the authors should state in 2.4 exactly when the lidocaine infusion was started and stopped
Thank you for your comment. We revised the time to start and stop the lidocaine infusion more exactly. (L109-110)
5) the most important problem with this study comes with the power calculation- the authors used a seemingly inappropriate study and comparision for this- Lap appys is not a good comparison as they are not elective and associated with significant preemptive pain. not to mention that the source of this paper is highly obscure. (possibly this data could be used to power a reasonable sample size)
We also very sorry about that. At the time of planning the study, it was difficult to find an adequate paper which described post-operative pain scoring FLACC scale in children undergoing any kinds of laparoscopic surgery. This was the almost only research which mentioned severity of pain as FLACC. Therefore, we had no choice but referred incidence of postoperative meaningful pain after laparoscopic surgery. In addition, our purpose was to demonstrate 30 % reduction of additional analgesics administration in PACU by lidocaine.
6) I dont think that with the subjective FLACC in pacu that a score difference of 3.8 to 5.3 has any clinical relevence- this is also bolstered by the fact that their treatment of pain was no difference.
Thank you for this comment. In terms of FLACC, there is no guideline for the minimal different score that signify clinical importance. Accordingly, it is debatable if this statistical difference correlates to clinical significance. Meanwhile, FLACC scale of 1 to 3 is considered mild discomfort, whereas 4 to 6 is considered moderate pain which should be treated. Therefore, it is still difficult to conclude that this difference is absolutely meaningless clinically. As your comment, we substituted the word “effectively” by “statistically”
7) section 4 page 6 ln 200- the comment on neuropathic pain is not relevant here and should be
removed.
Thank you for your comment. In addition to this statement, we have presented the following two references to refer to the appropriate capacity, but we also removed them, it could be not relevant.
References
Lee, B.; Koo, B.N.; Choi, Y.S.; Kil, H.K.; Kim, M.S.; Lee, J.H. Effect of caudal block using different volumes of local anaesthetic on optic nerve sheath diameter in children: a prospective, randomized trial. Br J Anaesth 2017, 118, 781-787, doi:10.1093/bja/aex078. Min, J.Y.; Lee, J.R.; Oh, J.T.; Kim, M.S.; Jun, E.K.; An, J. Ultrasonographic assessment of optic nerve sheath diameter during pediatric laparoscopy. Ultrasound Med Biol 2015, 41, 1241-1246, doi:10.1016/j.ultrasmedbio.2015.01.009.

Round 2
Reviewer 2 Report
The authors have not satisfactorily answered the main issues. 
1) the response to the issue surrounding the power analysis is flawed  one, there is adequate identification of peds pain in numerous populations.  This one study from a most obscure journal is a gross overestimate
2) the response to post operative analgesic plan is problematic. Simply stating that there was no control of analgesia post operatively is a fatal flaw in interpreting any post op data.